# Postpartum Spinal Cord Infarction: A Case Report and Review of the Literature

**DOI:** 10.3390/medicines9110054

**Published:** 2022-10-31

**Authors:** Jung-Lung Hsu, Shy-Chyi Chin, Ming-Huei Cheng, Yih-Ru Wu, Aileen Ro, Long-Sun Ro

**Affiliations:** 1Department of Neurology, New Taipei Municipal TuCheng Hospital, Chang Gung Memorial Hospital and Chang Gung University, New Taipei City 236, Taiwan; 2Department of Neurology, Chang Gung Memorial Hospital Linkou Medical Center and College of Medicine, Chang-Gung University, Taoyuan 333, Taiwan; 3Graduate Institute of Humanities in Medicine and Research Center for Brain and Consciousness, Shuang Ho Hospital, Taipei Medical University, Taipei 110, Taiwan; 4Department of Medical Imaging and Intervention, Linkou Medical Center, Chang Gung Memorial Hospital, Taoyuan 333, Taiwan; 5Division of Reconstructive Microsurgery, Department of Plastic and Reconstructive Surgery, Chang Gung Memorial Hospital, College of Medicine, Chang Gung University, Taoyuan 333, Taiwan; 6Section of Plastic Surgery, Department of Surgery, University of Michigan, Ann Arbor, MI 48105, USA; 7Department of Obstetrics and Gynecology, College of Medicine, Linkou Chang Gung Memorial Hospital and Chang-Gung University, Taoyuan 333, Taiwan

**Keywords:** postpartum, spinal cord, infarction

## Abstract

**Background**: Postpartum spinal cord infarction is a very rare disease. Only two cases have been reported in the English literature. **Methods**: We reported a 26 year old female who received second doses of the mRNA-1273 vaccine 52 days before delivery. She presented as sudden onset of paraplegia, sensory level, and sphincter incontinence at postpartum period. No history of heparin exposure was noted. Imaging findings confirmed the T10-11 level infarction and her anti–human heparin platelet factor 4 (anti-PF4) antibody was positive. After 7 days of dexamethasone therapy, her paraplegia and urinary incontinence gradually improved. **Results**: The CT angiography (CTA) of the artery of Adamkiewicz (Aka) showed tandem narrowing, most conspicuous at the T10-11 level, which was presumably due to partial occlusion of the arteriolar lumen. The thoracolumbar spine magnetic resonance imaging with contrast medium showed owl’s eyes sign at the T10 and T11 levels. We compared our case with two other case reports from the literature. **Conclusions**: Post-partum spinal cord infarction with positive anti-PF4 antibody and relatively thrombocytopenia are the characteristics of our case.

## 1. Introduction

Acute myelopathy is a neurological emergency characterized by a rapid progression of sensorimotor deficits with or without sphincter disturbances. The diagnosis of acute myelopathy presents a challenge to clinicians, as several possible etiologies, such as vasculature, demyelination/inflammation and compression, need to be considered in the differential diagnoses [1]. In these etiologies, spinal cord infarction is a devastating disease presenting as acute para- or quadriparesis with focal pain near the infarction site [2]. Most of the etiologies in the spinal cord infarction are due to cardiac or aortic surgery. The pathogenesis of spinal cord infarction was occlusion of the radicular artery or global hypoperfusion [3]. In the post-pandemic period, spinal cord involvement presenting as SARS-CoV-2 virus related acute transverse myelopathy, including spinal cord infarction, has also been reported [4].

Spinal cord infarction occurring at postpartum period is a very rare disease that has only been reported twice in the English literature [5,6]. The etiologies of postpartum spinal cord infarction include traditional vascular risk factors, aortic lesions and complications of epidural anesthesia. To further understand postpartum spinal cord infarction, we present a case and compare with the cases published in the literature.

## 2. Case Description

A 26-year-old female was sent to our hospital on her third postpartum day due to weakness and paresthesia in the bilateral lower extremities. This was her first uncomplicated pregnancy and an expected normal vaginal delivery. She had received epidural anesthesia at a local hospital at the L2-L3 regions. No heparin or anti-coagulant was used before or during the procedure. One and a half hours after delivery, she felt a sudden onset of bilateral lower extremity weakness and paresthesia below the L1 region. She also had micturition difficulty and constipation. Reviewing her past medical history, she received her first and second doses of the mRNA-1273 vaccine (Moderna) 83 and 52 days, respectively, before the onset of symptoms. She had no preceding illness, history of heparin exposure, neurological symptoms or collagen vascular disease. On examination, the blood pressure was 121/59 mm Hg and the pulse rate was 80 beats per minute. Mental status was normal and cranial nerve function was intact. There was a flaccid paraparesis (Medical Research Council Scale (MRC) score of her iliopsoas was 2 on the right side and 1 on the left side) and loss of bilateral knee and ankle reflexes. The Babinski sign was present on the right side and absent on the left side. A significant loss of pain and temperature sensation below the L1 level was found; however, vibration sensation and proprioception were intact. Clinical impression was spinal cord lesion below the thoracic level.

CT angiography (CTA) of the artery of Adamkiewicz (Aka) showed tandem narrowing, most conspicuous at the T10-11 level, which was presumably due to partial occlusion of the arteriolar lumen (Figure 1B). Thoracolumbar spine magnetic resonance imaging with contrast medium showed owl’s eyes sign at the T10 and T11 levels (Figure 1C). The spinal cord at the thoracolumbar region did not show any compression or stenosis. The platelet count was 222 × 10^9^ cells/L on admission, a reduction from a level of 250 × 10^9^ cells/L three days earlier. A high D-dimer concentration of 5213 ng/mL (reference range < 500 ng/mL) and a high FDP concentration of 15.8 µg/mL were also found (reference range < 5µg/mL) (Table 1). The sequential changes of fibrinogen levels were 3.21 mg/mL on day 4, 2.14 mg/mL on day 10, 2.21 mg/mL on day 14 and 3.51 mg/mL on day 18 (Reference range of fibrinogen level: 1.90–3.80 mg/mL). Her anti–human heparin platelet factor 4 (anti-PF4) antibody concentration was 58.89 ng/mL (reference range < 50) and no heparin was used before the testing. Spinal cord infarction with anterior spinal artery syndrome due to occlusion of the Aka was diagnosed. Under the impression of vasculitis that could not be totally excluded as a cause of spinal cord infarction, she received combination of intravenous dexamethasone 20 mg and antiplatelet medication (Cilostazol 100 mg) daily for seven days and shifted to oral prednisolone 30 mg on alternative days. Six days after admission, she could self-void after removal of the Foley catheter. On the fifth admission day, her MRC scores for the iliopsoas muscle gradually improved to 3 and 2 on the right and left sides, respectively. Her pain and temperature senses gradually improved. Her platelet counts gradually improved to 330 × 10^9^ cells/L, and her D-dimer levels gradually decreased to 2234 ng/mL on the 15th admission day (Figure 1A). She also received rehabilitation during the hospitalization periods and fortunately, after discharge, she could walk independently. 

### Review of the Literature

We conducted literature reviews of the three main databases for medical literature in English: Embase^®^, PubMed^®^, and MEDLINE^®^. We searched by combining keywords postpartum and spinal cord infarction: (“postpartum” or “peripartum”) AND (“spinal cord infarction”). The results from Embase, PubMed, and MEDLINE were merged to remove duplicates and underwent full-text review. There were a total 12 articles after removing duplicates. We excluded 11 non-spinal cord infarction and one non-human report. From this review of the literature, two cases of postpartum spinal cord infarction presenting as the anterior spinal cord syndrome at the T10-T11 level were documented. One case had normal coagulation profiles and another case had thrombocytosis with higher fibrinolytic activity [5,6]. A comparison between previous cases and our case is presented in Table 1.

## 3. Discussion

### 3.1. Incidence of Postpartum Stroke and Postpartum Spinal Cord Infarction

The causes of stroke occurring in association with pregnancy or puerperium/postpartum are not well understood. These uncertainties may contribute to the fear of compromise of treatment and etiological investigations of the mother and the unborn fetus. In one previous study, hemorrhagic strokes were more common than ischemic strokes, which are related to pregnancy [7]. Venous strokes are significantly more likely to occur postpartum compared with arterial strokes [8]. The risk of cerebral infarction is increased during the puerperium but not during pregnancy itself [9]. From French and US studies, the incidence of ischemic infarction was 4.3 per 100,000 deliveries and the relative risk of cerebral infarction was 0.7 during pregnancy, but it increased to 8.7 during the puerperium [10,11]. A recent study showed that the incidence of pregnancy-associated stroke was 14.5 per 100,000 deliveries. During the early postpartum period, incidence was five-fold greater compared to the first trimester [12]. Possible pathophysiological mechanisms that might contribute to ischemic strokes during gestation and puerperium include classic cardiovascular risk factors, alterations in coagulation status/hemodynamics, and pregnancy-specific disorders such as pre-eclampsia, eclampsia, choriocarcinoma or amniotic fluid embolism [9,13]. 

Particularly, spinal cord infarction is very rare in postpartum ischemic stroke. From a review of the literature, only two cases have been reported until now [5,6]. In 1981, Dunn reported a 26-year-old female presenting with sudden onset of deep burning pain below the chest, paraparesis and incontinence on her 20th postpartum day. The anterior spinal artery syndrome was diagnosed at that time. Hormonal effect on cerebral vascular endothelium or hyper-coagulopathy of pregnancy were suspected as possible etiologies [5]. In 2011, Soda et al. reported a 35-year-old woman presenting with acute onset, progressive low limbs paresthesia, weakness and dysuria for 3 days on her 6th postpartum day. The thrombocytosis and high fibrinolytic activity during the peripartum period were assumed to be a cause of anterior spinal artery syndrome in this case [6]. However, no anesthesia procedure, COVID or other vaccinations have been documented for both cases. In our case, the presentation of acute paraparesis with sensory dissociation and sphincter incontinence after normal vaginal delivery suggested postpartum anterior spinal cord infarction, which was confirmed by the imaging findings. Table 1 shows the comparison of laboratory data between previous literature and our case. A relatively lower platelet count with normal fibrinogen level was found during admission in our case, which is not compatible with the previous reports. 

### 3.2. Etiologies of Spinal Cord Infarction

Spinal cord infarction only accounts for 1.2% of all strokes and it is a rare disease that has devastating neurological sequelae to the patients [14]. The distribution of spinal cord infarction can be divided into the upper (cervical) segment and the lower (thoracolumbar) segment. The common etiologies of spinal cord infarction include atherosclerosis of the aorta, dissection of the aorta or aortic surgery, aortic or vessel dissection, systemic hypotension, spinal arteriovenous malformations and diving [3,15]. From our previous studies, patients with spinal cord infarction who had vessel dissection usually present with long-segment lesions (≥3 vertebral body spans) and posterior pattern involvement on the axial view compared to those without dissection [16]. Patients presenting with a long-segment lesion had poor 1-month outcomes compared to those presenting with a short-segment lesion. In this reported patient, her MRI showed a short-segment lesion and her outcome is quite good. Epidural anesthesia has been reported as one of the possible causes of spinal cord infarction [17]. One study of neurological complications after epidural and spinal anesthesia estimated that spinal cord infarction occurred in approximately five per million patients [18]. The pathophysiological mechanisms of spinal cord infarction following epidural anesthesia may contribute to intraoperative hypotension [19], toxicity from local anesthesia [20], or vasospasm related to the administration of a local epinephrine-containing anesthetic [21]. However, no clear associations have been demonstrated in each of the above cases. Another study found that, in elderly patients and those with abnormal spine anatomy, an excessive anesthetic agent volume may cause spinal cord compression [17]. In our case, CTA showed a tandem narrowing of the artery of Adamkiewicz (Aka) at the T10-11 level, which was far from the epidural anesthesia region (L2-L3 region) (Figure 2). Besides, neither the structural spine abnormality nor the complication of epidural procedure was found in our case. It is difficult to attribute the cause of spinal cord infarction to the epidural anesthesia related issues.

### 3.3. Characteristics of Our Postpartum Spinal Cord Infarction Case

From the above discussion, postpartum hyper-coagulopathy might be a common cause of spinal cord infarction. However, in this case, a particular finding was a positive test of anti-PF4 antibody after Moderna vaccination. Although increased anti-thrombin III and protein C levels were also occurring in our case, these might be related to the coagulation disorder during the postpartum period. Increased anti-thrombin III level could be found in anabolic steroid usage, hemophilia or low level of vitamin K [22]. Increased plasma protein C level with decreased anti-thrombin III level could also be found in nephrotic syndrome [23]. Thus, hyper-coagulopathy of pregnancy may play a role in the spinal cord infarction in our case. Only trace levels (<20 ng/mL) of PF4 are normally found in plasma, but rapidly increase as much as 30-fold upon heparin administration, primarily by heparin displacing PF4 from the endothelial cells’ surface [24]. In our case, neither the history of heparin exposure nor the preceding medical illness, hematological disorders or collagen vascular disease were found. Recently, vaccine-induced thrombosis with thrombocytopenia (VITT) has been reported after vaccination with recombinant adenovirus vectors (ChAdOx1 vaccine from AstraZeneca and the Ad26.COV2.S vaccine from Johnson & Johnson/Janssen) [25,26]. Thrombi have been documented in the cerebral veins, splanchnic veins, pulmonary veins and other arteries [27]. The presence of anti-PF4 antibody (sensitivity 97%, specificity 92% at our laboratory using enzyme-linked immunosorbent assay (ELISA)) [28], high concentration of D-dimer and lower platelet count than on the 11th admission day might suggest that the patient had thrombosis with thrombocytopenia syndrome [29]. Although the platelet count of our patient did not decrease to less than 150 × 10^9^/L or 50% less from the baseline platelet count, several case reports of VITT with a platelet count greater than 150 × 10^9^/L have been documented [27,30]. The interval from vaccination to symptom onset usually ranges between 4–28 days [31]; however, cases occurring between 30–100 days have also been documented [32]. In addition, a search from the Vaccine Adverse Event Reporting System (VAERS) created by the American Food and Drug Administration (FDA) and Centers for Disease Control and Prevention (CDC) using the criteria of COVID19 vaccine, cerebral infarction and onset interval showed two cases with an onset interval longer than 30 days after vaccination (https://vaers.hhs.gov/data.html accessed on 12 February 2022). Most vaccine-induced thrombosis with thrombocytopenia (VITT) cases were associated with two kinds of vaccines based on recombinant adenovirus vectors [26]. Recently, VITT from post-Moderna vaccination has also been reported [33]. Although our case had a longer time interval between onset of symptoms and vaccination that does not perfectly meet the possible criteria of VITT, we still could not totally exclude the possibility of VITT in postpartum spinal cord infarction after Moderna vaccination [32,34]. 

### 3.4. Limitations

There are some limitations in our study. First, we did not perform a conventional spinal cord angiography study to confirm the anterior spinal cord infarction in our case, but CTA studies did show tandem narrowing of the artery of Adamkiewicz (Aka) at the T10-11 level. However, previous studies showed that recently developed CTA may efficiently display the anatomical features and comprehensively evaluate the injury patterns of the anterior spinal artery or Aka [35,36]. Nevertheless, conventional spinal cord angiography remains the gold-standard diagnosis of spinal cord infarction. Second, we only used the ELISA method to detect anti-PF4 antibody. We did not perform a sophisticated serotonin releasing assay because the assay is not available at our hospital. We acknowledge that using the anti-PF4 antibody ELISA test to screen the possibility of VITT and further performing the platelet serotonin-release assay to confirm VITT is a more precise methodology [28,37]. Last, we did not have the baseline platelet counts for this patient so that we could not meet the criteria of decreasing more than 50% from the baseline platelet count in VITT. A longitudinal clinical follow-up of the platelet counts in this patient is warranted.

## 4. Conclusions

In summary, postpartum spinal cord infarction is an unusual disorder. Epidural anesthesia, local abnormalities in spinal structure and vascular injury could be the possible etiologies. Furthermore, we should be aware of the possible adverse events of vaccination, given the current use of recombinant adenovirus vectors and mRNA vaccination against SARS-CoV-2. Currently, close monitoring for thromboembolic outcomes with thrombocytopenia after vaccination with all COVID-19 vaccines will be needed [38]. Nevertheless, our findings in this case need to be confirmed by further additional reports or laboratory data but should not hamper persons from receiving COVID-19 vaccines because of the very rare incidence of postpartum spinal cord infarction after mRNA-1273 vaccination. The benefit to risk ratio in receiving COVID-19 vaccines is quite apparent. 

## Figures and Tables

**Figure 1 medicines-09-00054-f001:**
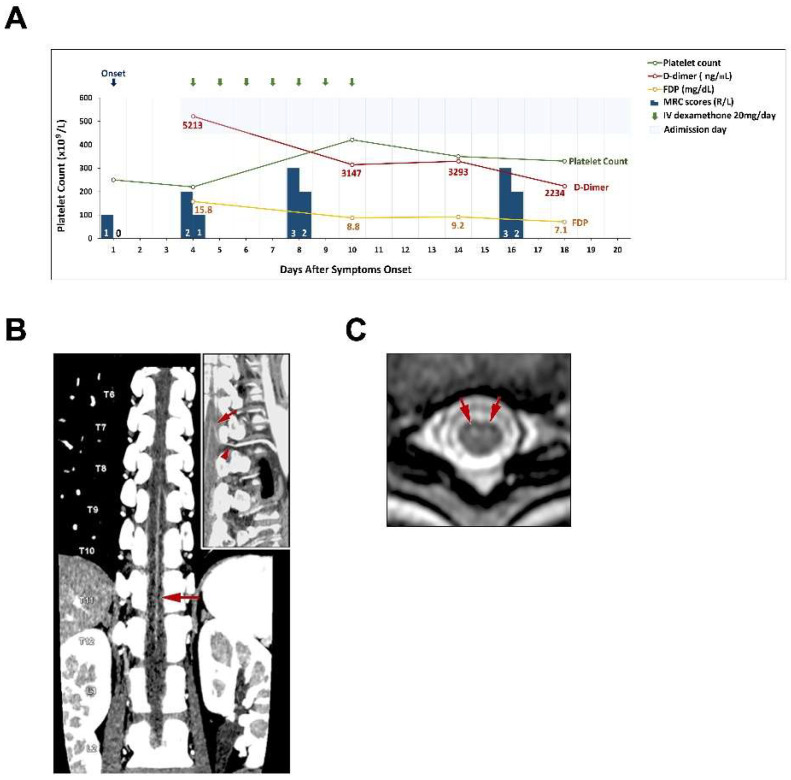
Clinical course and spinal cord imaging in this case. (**A**). Summary of clinical events and laboratory studies are presented. (**B**). Maximum intensity projection reconstruction of a coronal CTA image reveals tandem narrowing of the Adamkiewicz artery, more severe at the T10-11 level (arrow); the insert image is a curved multiplanar reformatted image that shows a single Adamkiewicz artery (arrow) with optimal continuity around the pedicle of the vertebral arch (arrowhead) at the left T9-10 lumbar artery and aorta. (**C**). T2-weighted axial image shows focal edema in the central cord gray matter at the T9-11 levels with greater involvement of the ventral horns (owl’s eyes sign, arrows).

**Figure 2 medicines-09-00054-f002:**
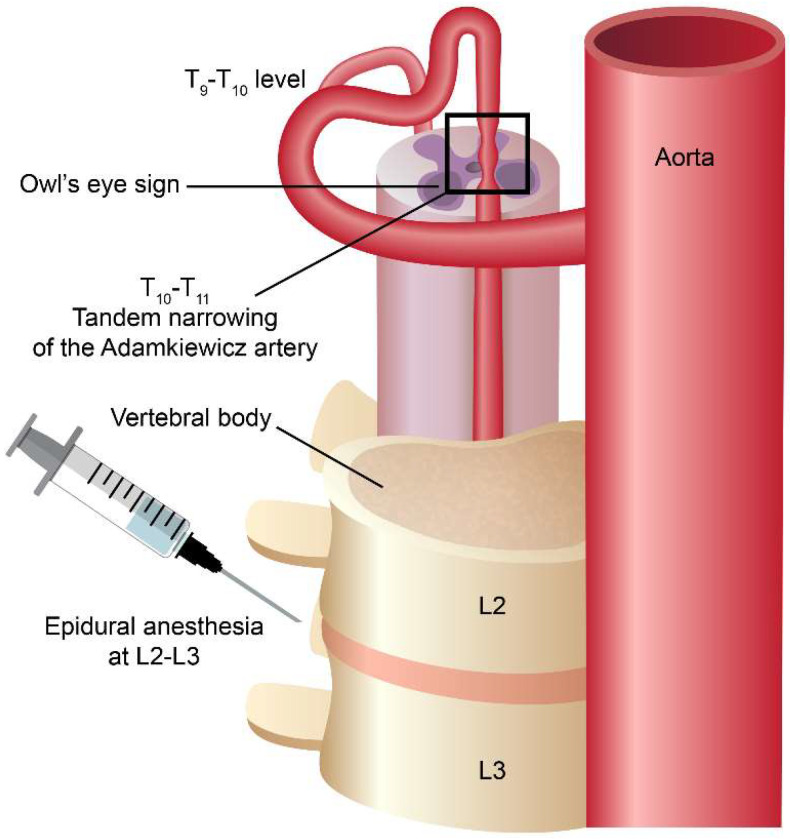
The cartoon figure depicts the spatial relationship between the epidural anesthesia site and the spinal cord infarction site. The infarction location is a little far from the epidural anesthesia location, thus the epidural anesthesia itself as the possible cause of the spinal cord infarction is less likely.

**Table 1 medicines-09-00054-t001:** Results from the selected laboratory data in our case and review of the literature.

Study, Units	Dunn DW et al. Case	Soda T et al. Case	Our Case (Reference Value)
Polymerase chain reaction test for SARS-CoV-2 virus using a nasopharyngeal swab	NA	NA	negative (negative)
Initial platelet count, ×10^9^ cells/L	NA	623 *	222 (150–400)
fibrinogen level, mg/mL	NA	4.66 *	3.21 (1.90–3.80)
Fibrin degradation products, μg/mL	NA	10.2	15.8 (<5)
Peak D-dimer, ng/mL	NA	5200	5213 (<500)
Peak activated partial thromboplastin time, s	normal	normal	23.4 (24.6–33.8)
Prothrombin time, s	normal	normal	11.1 (8–12)
Anti-human heparin platelet factor 4 antibody, ng/mL	NA	NA	58.89 * (<50)
Anticardiolipin antibody level			
IgG GPL-U/mL	NA	NA	5.8 (<10)
IgM GPL-U/mL	NA	NA	5.7 (<10)
Protein C level, %	NA	normal	>150 * (70–140)
Protein S level, %	NA	normal	64 (60–140)
Antithrombin III level, %	NA	NA	146 * (75–125)
Antinuclear antibodies	negative	negative	negative
VDRL	negative	negative	negative
Lupus anticoagulant test	NA	NA	negative (negative)
Procalcitonin, ng/mL	NA	NA	<0.02 (<0.02)

s: second. NA: not available. VDRL: venereal disease research laboratory. *: higher than reference value.

## Data Availability

The datasets presented in this article are not readily available. Please contact corresponding author Long-Sun Ro Email: cgrols@adm.cgmh.org.tw if you are interesting about this case.

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
