# Peer review of "Postpartum Spinal Cord Infarction: A Case Report and Review of the Literature"

_medicines, 2022, doi:10.3390/medicines9110054_

Round 1
Reviewer 1 Report
Thank you for presenting an rare case of spinal cord infarction postpartum. I have a couple of questions/comments:
1. Could you address why was thrombolysis not considered as this would be standard of care for stroke. Was it that the local providers did not recognize that she had suffered from a cord infarct or was it due to concern about the bleeding related to the epidural anesthesia?
2. Steroids are not generally used to treat ischemic stroke especially when there is no evidence of significant swelling. Why did you chose to use steroids and what is the evidence that it works?
As regards etiology I agree that neuraxial anesthesia was not the culprit. I do think that both VITT and the hypercoaguability of pregnancy played a role as presumably the anti-PF4 antibodies were there prior to delivery. Thus I am not sure I agree with the statements in lines 172-176.
Author Response
We extend our gratitude to the reviewers who provided kind and constructive comments.
Reviewer 1
- Could you address why was thrombolysis not considered as this would be standard of care for stroke. Was it that the local providers did not recognize that she had suffered from a cord infarct or was it due to concern about the bleeding related to the epidural anesthesia?
A: Thanks for reviewer’s inquiry and it is an important issue regarding the strategy of treatments. Indeed, the thrombolysis therapy is a stand of care for stroke patients. However, this patient was sent to our hospital with bilateral lower extremity weakness and paresthesia on her third postpartum day that was out of the golden hours of recombinant tissue-type plasminogen activator (rt-PA, within 4.5 hours) and intra-arterial thrombectomy (IAT, within 24 hours) usage. Although she had received epidural anesthesia for normal vaginal delivery at a local hospital, she also had massive postpartum hemorrhage at that time, which rendered her not suitable for thrombolytic therapy during that time. We rephrased our sentence as: A 26-year-old female was sent to our hospital on her third postpartum day due to the weakness and paresthesia in the bilateral lower extremities.
- Steroids are not generally used to treat ischemic stroke especially when there is no evidence of significant swelling. Why did you choose to use steroids and what is the evidence that it works?
A: We totally agree with reviewer’s comments. This patient was a healthy lady previously and she didn’t have cardiovascular risk factors such as, unhealthy diet, physical inactivity, type 2 diabetes mellitus, hypertension, dyslipidemia, tobacco use and harmful use of alcohol. On the CT angiography (CTA) of the artery of Adamkiewicz (Aka), the report demonstrated tandem narrowing, which could not exclude the possibility of vasculitis. Thus, we utilized steroid combined with anti-platelet medication (Cilostazol) for initial treatment. From review of the literature, glucocorticoids and cyclophosphamide are most often used in CNS vasculitis for induction therapy (Rev Neurol (Paris). 2022 Sep 22; S0035-3787(22)00718-4.). We rephrased as: Under the impression of vasculitis that could not be totally excluded as a cause of spinal cord infarction, she received combination of intravenous dexamethasone 20 mg and antiplatelet medication (Cilostazol 100 mg) daily treatment for seven days and shifted to oral prednisolone 30 mg on alternative days.
As regards etiology I agree that neuraxial anesthesia was not the culprit. I do think that both VITT and the hypercoaguability of pregnancy played a role as presumably the anti-PF4 antibodies were there prior to delivery. Thus, I am not sure I agree with the statements in lines 172-176.
A: We thank reviewer’s comments on the regarding issue. We rephrased as: From above discussion, postpartum hyper-coagulopathy might be the common cause of spinal cord infarction. However, in this case, a particular finding was a positive test of anti-PF4 antibody after Moderna vaccination. Although increased anti-thrombin III and protein C levels were also occurring in our case, these might be related to the coagulation disorder during the postpartum period. Increased anti-thrombin III level could be found in anabolic steroid usage, hemophilia or Low level of vitamin K [22]. Increased plasma protein C level with decreased anti-thrombin III level could also be found in nephrotic syndrome [23]. Thus, hyper-coagulopathy of pregnancy may play a role in the spinal cord infarction in our case.
Reviewer 2 Report
The manuscript is well written with some minor modifications and clarification needed to be done in below.
1. Line 54-56. Please clarify if the symptom onset is on the third day postpartum or 1.5 hours after delivery.
2. Line 100. Grammer mistake. "was presented". There are also minor grammar mistakes throughout the manuscript. PLease review carefully.
3. Line 131-133. Please summarize with paragraph or table about both case reports presented in the manuscript. (eg. onset, symptoms, location, anesthesia method, COVID vaccination status, etc.)
4. Please present the trend of fibrinogen if available..
Author Response
We extend our gratitude to the reviewers who provided kind and constructive comments.
Reviewer 2
- Line 54-56. Please clarify if the symptom onset is on the third day postpartum or 1.5 hours after delivery.
A: Thanks for reviewer’s comments. Her symptom onset was 1.5 hours after delivery. We correct our descriptions as: A 26-year-old female was sent to our hospital on her third postpartum day due to the weakness and paresthesia in the bilateral lower extremities.
- Line 100. Grammer mistake. "was presented". There are also minor grammar mistakes throughout the manuscript. PLease review carefully.
A: Thanks for that reviewer carefully checked up our manuscript. We then corrected Line 100 grammar mistake.
- Line 131-133. Please summarize with paragraph or table about both case reports presented in the manuscript. (eg. onset, symptoms, location, anesthesia method, COVID vaccination status, etc.)
A: Thanks for reviewer’s suggestions. We have added a section to Line 131-133: In 1981, Dunn reported a 26-year-old female presented as sudden onset of deep burning pain below the chest, paraparesis and incontinence on her 20th postpartum day. The anterior spinal artery syndrome was diagnosed at that time. Hormonal effect on cerebral vascular endothelium or hyper-coagulopathy of pregnancy were suspected as possible etiologies [5]. In 2011, Soda et al. reported a 35-year-old woman presented as acute onset, progressive low limbs paresthesia, weakness and dysuria for 3 days on her 6th postpartum day. The thrombocytosis and high fibrinolytic activity during the peripartum period were assumed to be a cause of anterior spinal artery syndrome in the case [6]. However, no anesthesia procedure, COVID or other vaccinations have been documented for both cases.
- Please present the trend of fibrinogen if available.
A: Thanks for reviewer’s suggestions. The sequential changes of fibrinogen levels were 3.21 mg/mL on day 4, 2.14 mg/mL on day 10, 2.21 mg/mL on day 14 and 3.51 mg/mL on day 18. Our reference range of fibrinogen level is 1.90-3.80 mg/mL. We have added above values to the main text.
Reviewer 3 Report
thank you for this excellent review, I do not have any suggestion for further improvement
Author Response
Reviewer 3
thank you for this excellent review, I do not have any suggestion for further improvement
A: Thanks for reviewer’s positive comments on our manuscript.